# The Genetic Mutation of *ANO5* in Rabbits Recapitulates Human Cardiomyopathy

**Tingting Sui, Hongwu Yao, Tao Zhang, Jinze Li, Liangxue Lai * and Zhanjun Li ***

Animal Embryo Engineering Laboratory, Institute of Zoonosis, Jilin University, Changchun 130062, China; suitingting@jlu.edu.cn (T.S.); yaohw19@mails.jlu.edu.cn (H.Y.); ztao19@mails.jlu.edu.cn (T.Z.); ljz98_jlu@163.com (J.L.)

* Correspondence: lailx@jlu.edu.cn (L.L.); lizj_1998@jlu.edu.cn (Z.L.)

**Abstract:** The limb girdle muscular dystrophy type 2L (LGMD2L) is caused by mutations of the *ANO5* gene in humans which encodes a 913 amino-acid integral membrane protein. Although cardiomyopathy has been reported in patients with an *ANO5* mutation, the *ANO5* mutant mice did not recapitulate this phenotype in previous studies. This study demonstrated that the $ANO5^{-/-}$ rabbits recapitulated the typical signs of cardiomyopathy with decreased ejection fraction (EF) and fraction shortening (FS) with increased interstitial fibrosis. This $ANO5^{-/-}$ rabbit model would promote basic research to comprehend the pathogenesis and mechanism of *ANO5*-related cardiomyopathy.

**Keywords:** CRISPR/Cas9; anoctamin 5; cardiomyopathy; rabbit

## 1. Introduction

The *ANO5* gene, which encodes a 913 amino acid integral membrane protein, is highly expressed in skeletal muscle, cardiac muscle, and bone tissues [1–4]. It has been reported that the recessive mutations of *ANO5* ($ANO5^{-/-}$) are associated with limb-girdle muscular dystrophy 2L (LGMD2L), the characteristics of which resemble that of dysferlinopathies; gnathodiaphyseal dysplasia, bone fragility, cortical thickening, and sclerosis of tubular bone diaphysis; and also cardiomyopathy in previous studies [5–7].

Ten anoctamin family members have been identified so far with important functions in various physiological process [8–10]. Due to the structural similarity between the *ANO5* gene and other anoctamin proteins, it was proposed that *ANO5* may have function of $Ca^{2+}$-activated $Cl^-$ channels (CaCC). Recently, some anoctamins, including *ANO1*, *ANO2*, *ANO8*, and *ANO9*, have been linked to the regulation of CaCC activities; however, other anoctamins, such as *ANO3* and *ANO7*, which are short on CaCC activities, have localization in cellular activities [6,11,12]. This suggests that the anoctamin family could possess different functional properties among different family members and also that the molecular and cellular mechanisms of the *ANO5* gene still remains unclear.

Previous studies have reported that *ANO5* knockout (KO) mice have different pathological manifestations. The *ANO5* gene trapped mouse model shows signs of muscular dystrophy [13] without any obvious cardiomyopathy [14], reminiscent of the phenotype of LGMD2L patients. Currently, there are no ideal animal models to study the cardiomyopathy phenotype reported in *ANO5*-mutated patients, and also the cellular functional studies of the *ANO5* function in skeletal muscle and myocardium still need to be further confirmed. Moreover, rabbits may be more popular animal models compared to mice in simulating some human diseases and have higher similarity with human beings in some respects, such as genetics, physiology, and anatomy, compared to mice [15].

Therefore, the phenotype of cardiomyopathy was studied using $ANO5^{-/-}$ rabbits generated in our previous study [16]. Our data demonstrate that the typical phenotype of cardiomyopathy was identified in this $ANO5^{-/-}$ rabbit model for the first time.

## 2. Materials and Methods

### 2.1. Animal and Ethics Statement

The *ANO5* gene editing rabbits were generated by the CRISPR/Cas9 system in our laboratory [16] and kept at the Jilin University Laboratory Animal Centre.

### 2.2. Genotyping of Rabbits

Genotyping of *ANO5* KO rabbits was performed according to the manufacturer's instructions as described previously [17,18]. Briefly, the exon of the *ANO5* gene was amplified by PCR and Sanger sequenced for rabbits' mutations. The obtained sequences were compared with the corresponding reference sequence and analyzed using Snapgene (NC_013669.1). The same set of PCR primer was used as in our previous study [16].

### 2.3. Histology Analysis

The tissue sample of cardiac muscle was collected from $ANO5^{+/-}$, $ANO5^{-/-}$, and wild-type (WT) rabbits (15 months of age). Histology analysis was performed as per a previous study [16]. The 5 μm sections for HE, Masson's trichrome, and Van Gieson staining were performed as described previously [19,20]. The pictures of stained sections were captured by a Nikon TS100 microscope.

### 2.4. Echocardiography

Echocardiography was carried out as described previously [21,22]. The cardiac dimensions left ventricular end-diastolic diameter (LVDd), the percentage of fractional shortening (FS), and left ventricular ejection fraction (EF) were determined.

### 2.5. Statistical Analysis

The data were statistically analyzed by GraphPad prism 8.0.2 (*t*-test), and $p < 0.05$ was used as statistically significant, * $p < 0.05$.

## 3. Results

### 3.1. The Breeding and Genotyping of ANO5 Gene-Edited Rabbits

The *ANO5* KO rabbits were generated in our previous study [16]. To breed *ANO5* gene-edited rabbits, the heterozygous male was mated with heterozygous female *ANO5* gene-edited rabbit (Figure S1A). The genotyping PCR result showed that 4 $ANO5^{+/-}$ and 6 $ANO5^{-/-}$ pups were generated in this study (Figure S1B).

### 3.2. Pathological Changes of Cardiac in the ANO5$^{-/-}$ Rabbits

To determine whether the disruption of *ANO5* in rabbits induces the typical phenotype of cardiomyopathy, the histological and functional cardiac changes were evaluated and compared between gene-edited *ANO5* ($ANO5^{+/-}$ and $ANO5^{-/-}$) and WT rabbits. The results showed that the increased interstitial fibrosis was determined in $ANO5^{-/-}$ rabbits, but there was no significant difference in $ANO5^{+/-}$ rabbits compared to WT control at the age of 15 months (Figure 1A); 6 month old $ANO5^{-/-}$ rabbits did not show significant differences compared to WT control (Figure S2). In addition, the echocardiography result showed the significantly increased diastolic diameter of the left ventricle (LVDD), while EF and FS decreased in the $ANO5^{-/-}$ rabbits, compared to the WT controls (Figure 1B–E).

These data suggest that the typical phenotype of cardiomyopathy was identified in the *ANO5*$^{-/-}$ rabbit models.

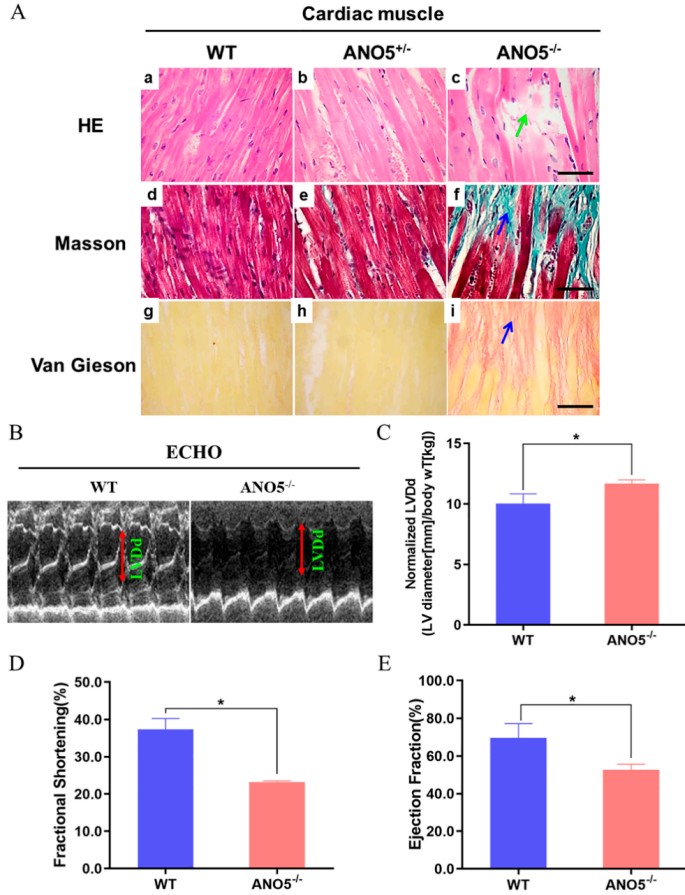

**Figure 1.** (**A**) H&E staining, Masson' trichrome, and Van Gieson staining of cardiac muscles from wild-type (WT), *ANO5*$^{+/-}$ and *ANO5*$^{-/-}$ rabbits. (**B**) Representative echocardiograms of *ANO5*$^{-/-}$ and WT control. The red arrows represent the position of left ventricular end-diastolic diameter (LVDD). (**C**) The left ventricular diastolic diameter increased in *ANO5*$^{-/-}$ rabbits. (**D**) The fractional shortening decreased in *ANO5*$^{-/-}$ rabbits. (**E**) The left ventricular ejection fraction (EF) decreased in *ANO5*$^{-/-}$ rabbits. The data is shown by means ± SEM and was analyzed by *t*-tests. * $p < 0.05$. Normalized LVDD, The ratio of LV diastolic diameter to body weight.

## 4. Discussion

The high expression of the *ANO5* gene in human skeletal muscle, heart muscle, and bone [1–3]. Together with cardiomyopathy was reported in *ANO5*-deficient patients [7,23]. In our study, the *ANO5*$^{-/-}$ rabbits exhibited cardiomyopathy at 15 months as demonstrated by histological and functional cardiac analysis, while there is no obvious cardiomyopathy with normal thickness of the interventricular septum and cardiac functions in the *ANO5*$^{-/-}$ mouse [13,14]. The 15 month old *ANO5*$^{-/-}$ rabbits showed cardiac changes close to *ANO5*-deficiency patients (age 20–50) [24] and mice (age 6 months or later) [13] with the only symptoms muscular dystrophy typically appearing. Interestingly, the function of cardiac appeared abnormal by reduced left ventricular EF and FS. However, in some patients with cardiac arrhythmia [24], while the echocardiogram displayed LV dilatation and LV dysfunction as seen in other patients, but dilated cardiomyopathy may be a complication in muscles of *ANO5*-deficient patients [7].

There are some differences between mice and rabbits, which may trigger the performance of some aspects of the genetic disorder. The type of natural mutations is crucial for different consequences in animal models of different species. The rabbits, with a small indel in exon 12 were used to perform the

determination of cardiac pathology. The CRISPR-induced indels within the exon 12 or 13 of the *ANO5* gene lead to the development of pathological alterations in various muscles and cardiac changes of the rabbit, resembling human patients with *ANO5* mutations [25,26]. In contrast, mice with exon 1 or exon 2 deletion did not develop muscle and cardiac phenotypes [14] which may illustrate the different consequences of cardiac pathology in *ANO5* mutant models.

To our knowledge, this novel animal model which recapitulates human cardiomyopathy will be beneficial for studying the potential impact of the disruption of *ANO5* on cardiac changes and would also promote understanding of the pathogenesis mechanism of *ANO5*-related cardiomyopathy in the future.

**Supplementary Materials:** The following are available online at http://www.mdpi.com/2076-3417/10/14/4976/s1. Figure S1: The generation of F1 carrying ANO5 mutant rabbits; Figure S2: The histological of WT and *ANO5*$^{-/-}$ at the age of 6 months.

**Author Contributions:** Conceptualization, S.T. and Z.L.; methodology, H.Y.; software, T.Z.; validation, J.L., H.Y. and L.L.; formal analysis, T.S.; investigation, T.Z.; resources, H.Y.; data curation, J.L.; writing–original draft preparation, T.S.; writing–review and editing, Z.L.; visualization, H.Y.; supervision, L.L.; project administration, Z.L.; funding acquisition, Z.L. All authors have read and agreed to the published version of the manuscript.

**Funding:** This research was funded by the Changjiang Scholars and Innovative Research Team (no. IRT_16R32) and the China Postdoctoral Science Foundation (grant nos. 2019TQ0113 and 2020M670859).

**Acknowledgments:** We thank embryo manipulation technicians for the technical support.

**Conflicts of Interest:** The authors declare no conflict of interest.

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
