# Peer review of "The Genetic Mutation of ANO5 in Rabbits Recapitulates Human Cardiomyopathy"

_applsci, doi:10.3390/app10144976_

Round 1
Reviewer 1 Report
In this brief report by Sui et al., author showed that the genetic deficient ANO5 rabbit recapitulate the cardiomyopathy phenotype which present in Limb girdle muscular dystrophy type 2L (LGMD2L) patients. Author claimed that this phenotype was not shown by ANO5 deficient mouse model. This is an interesting model to study the cardiomyopathy pathogenies associated with LGMD2L. In order to improve the manuscript authors need to address following comments.
- Did authors looked the expression of ANO5 gene in Wt, ANO5+/- and ANO5-/- tissue at both protein and mRNA level to make sure that this is ANO5 deficient model.
- Describe the rational of targeting exon 12-13 to generate ANO5 deficient rabbit.
- Does mice with similar KO strategy (targeting exon 12-13) is available? If available what is the status of cardiomyopathy phenotype.
Reviewer 2 Report
This paper is important from ANO5 function better understanding. And even, it is a very interesting new tool to work on the ethiopathogenic view of LGMD2L and future treatments for this disease.
Introduction is quite adequate, but you should include the different symptoms and signs described in ANO5 positive patients and, for sure, its relevance during the disease. Morover, you should refer to the symptoms and signs described in the mouse knockout animal, and its relation with human disease.
About the methodology, the genetic profile obtained with CRISPR/Cas9 technology in your rabbits should be represented comparing to the knockout mice studied before.
Echocardio studies are properly curried out, and the results prove enough differences with mice model.
In the discussion, you should evaluate the importance of cardiac symptoms appearing at old stages of rabbits (15 months) instead of earlier stages. Comparing the last with both the mice and the human disase.
Finally, there are mistakes using language throughout all the paper.
Reviewer 3 Report
Manuscript with title - The genetic mutation of ANO5 in rabbits recapitulates human cardiomyopathy.
The authors of the manuscript focused on the basic research to comprehend the pathogenesis and mechanism of ANO5 related cardiomyopathy using an animal model
The methodology refers to previously published manuscripts. The authors should further describe the methodology of Sanger sequencing used in genetic analysis of is described in the manuscript:,,Simurda T, et al. Int J Mol Sci. 2017 Dec 29;19(1). pii: E100. doi: 10.3390/ijms19010100“. The authors should add this citation to the manuscript.
The whole manuscript is brief but transparent and is filled with new knowledge that is important in the research of internal medicine. Figure in the text ide very clearly written.
I have to say that with these 23 references there are only 10 references that are more than 5 years old. What confirms that this article is important to this field of medicine.
The authors must correct the methodological part, after accepting the above requirements, the manuscript may be published.
Round 2
Reviewer 1 Report
Authors addressed the issue raised by the reviewer and revised the manuscript accordingly.
Author Response
Dear reviewer,
Thank you very much for the thorough and thoughtful comments.
Thank you again for your email and kindly suggestions. We would try our best to edit the manuscript to fit for the requirement of Applied sciences.
Sincerely Yours,
Zhanjun Li
Reviewer 3 Report
The presented manuscript has been corrected in response to the suggestions. The authors have followed the recommendations of the reviewers. After the revision, the provided data and interpretation of the results became more clear. I would like to thank the authors for resubmitting the manuscript and explaining the obscure points from the previous version. Now, the revised manuscript can be accepted for publication.
Author Response

(The authors gave the same response as above.)
